# Anti-Inflammatory Food in Asthma Prepared from Combination of *Raphanus sativus* L., *Allium hookeri*, *Acanthopanax sessiliflorum*, and *Dendropanax morbiferus* Extracts via Bioassay-Guided Selection

**DOI:** 10.3390/foods11131910

**Published:** 2022-06-27

**Authors:** Kyung-Dong Lee, Sun-Yup Shim

**Affiliations:** 1Department of Companion Animal Industry, College of Health & Welfare, Dongshin University, Naju 58245, Korea; leekd@dsu.ac.kr; 2Department of Food Science and Technology, College of Life Science and Natural Resources, Sunchon National University, Suncheon 57922, Korea

**Keywords:** herbal plant, anti-inflammatory food, hepatoprotective, asthma, air pollution

## Abstract

Asthma is a highly prevalent inflammatory disease of the respiratory airways and an increasing health risk worldwide. Hence, finding new strategies to control or attenuate this condition is necessary. This study suggests nutraceuticals that are a combination of herbal plant extracts prepared from *Acanthopanax sessiliflorum* (AS), *Codonopsis lanceolate* (CL)*, Dendropanax morbiferus* (DM), *Allium hookeri* (AH), and *Raphanus sativus* L. (RS) that can improve immunomodulatory ability through the detoxification and diuresis of air pollutants. Herbal parts (AH whole plant, RS and CL roots, AS and DM stems, and DM leaves) were selected, and four types of mixtures using plant extracts were prepared. Among these mixtures, M2 and M4 exhibited antioxidant activities in potent 2,2-azino-bis (3-ethylbenzothiazoline-6-sulfonic acid) diammonium salt (ABTS) and 1,1-diphenyl-β-picrylhydrazine (DPPH) radical assays. Moreover, M4 exhibited a marked increase in glutathione S-transferase (GST) activity and significantly inhibited the inflammatory mediator, nitric oxide (NO) and proinflammatory cytokines, interleukin (IL)-1β, IL-6 and tumor necrosis factor (TNF)-α generation. Furthermore, M4 exhibited the strongest antioxidant, hepatoprotective, and anti-inflammatory effects and was selected to prepare the product. Before manufacturing the product, we determined that the active mixture, M4, inhibited gene expression and generation of proinflammatory cytokines IL-1β, IL-6 and TNF-α in ovalbumin (OVA)-, lipopolysaccharide (LPS)-, and particulate matter (PM)-induced asthmatic rat models. The granular product (GP) was manufactured using M4 along with additives, i.e., lactose, oligosaccharide, stevioside extract, and nutmeg seed essential oils (flavor masking), in a ratio of 1:4 using a granulation machine, dried and ultimately packaged. The GP inhibited the generation of proinflammatory cytokines IL-1β, IL-6 and TNF-α in OVA-, LPS- and PM-induced asthmatic rat models. These results suggest that GP prepared from a combination of herbal plants (AS, CL, DM, AH and RS) is a potent functional food with anti-inflammatory activity that can be used to treat asthma caused by ambient air pollutants.

## 1. Introduction

Inflammation is a complicated physiological response of body tissues to detrimental stimuli and may result in inflammatory disorders such as allergies, asthma, autoimmune diseases, coeliac disease, hepatitis, inflammatory bowel disease, and reperfusion injury [1,2,3,4]. Asthma, a chronic condition of the respiratoty airway, causes airway disturbance and bronchial hyper-responsiveness, accompanied by symptoms involving coughing, wheezing, shortness of breath, chest tightness, and pain [5,6,7,8]. Allergic asthma is prolonged lung inflammation caused by the permeation of eosinophils, leukocytes and neutrophils into the alveoli as demonstrated by bronchoalveolar lavage (BAL). It is a common inflammatory lung disease triggered by animal dander, dust mites, non-steroidal anti-inflammatory drugs, mold, pollen, respiratory infections and fine dust [1,2,3,4]. Ambient air pollution is a major environmental threat to urban populations and has crucial health effects such as diminished life expectancy, increased daily mortality and hospital admissions, adverse birth outcomes and asthma progression [9]. It is composed of gaseous constituents and airborne particulate matter (PM), classified based on particle size as coarse particles of 10 μm or smaller (PM 10), fine particles (PM 2.5) and ultrafine particles (PM 0.1) [10]. PM is naturally inhaled when suspended in the air and reaches the lower respiratory tract, accumulating in the distal conducting airways [10]. Ultimately, PM is deposited in these airways, interacts with the airway cells and causes an inflammatory immune response. There is considerable interest in natural resources that can ameliorate various diseases caused by ambient air pollution [11,12]. Natural products have made extensive contributions to human health in drug discovery and functional food development. The search for new nutraceuticals to protect and treat inflammatory diseases is critical. Herbal medicines play a consequential role in inflammatory disorders. Moreover, combined herbal therapy extracts are superior to individual herbal treatments due to the synergistic effects of their constituents and the neutralization of the possible toxicity and side effects of each of the individual plants [13,14]. *Acanthopanax sessiliflorum* (AS), belonging to the family Araliaceae, is one of the most abundant species in the genus *Acanthopanax* and has been used in traditional medicine to treat diseases, such as rheumatoid arthritis, diabetes, bacterial infections, cancer, inflammation and hypertension. AS contains various bioactive components involving saponins, which have anti-inflammatory, antioxidant and antidiabetic effects [15,16,17]. *Codonopsis lanceolate* (CL), belonging to the family Campanulaceae, is a perennial herb known as the bonnet bell flower and is traditionally used as a vegetable as well as a medicinal plant. The roots of this plant have been used to treat asthma, bronchitis, cancer, cough, dyspepsia, hyperlipidemia, obesity, psychoneurosis and tuberculosis. For a long time, the pharmacological effects of CL, including anticancer, anti-inflammatory, antimicrobial, antioxidant and immunomodulatory activities, have provoked interest [18]. *Dendropanax morbiferus* (DM), belonging to the family Araliaceae, has been used as an alternative medicine for lung and neurological diseases, such as paralysis, stroke and migraine. This plant has various biochemical properties including anti-inflammatory, antioxidant, anticancer, hepatoprotective and antidiabetic effects. AS, CL and DM contain various physiological bioactive compounds with cholesterol lowering, anti-inflammatory, antioxidant, antileishmanial and anticancer properties [19,20,21].

*Allium hookeri* (AH), a plant of the family Allium, which includes garlic, chives and onion, has various physiological effects including antidiabetic, antiobesitic and antiadipogenetic effects [22,23,24].

*Raphanus sativus* L. (RS), a plant of the family Cruciferae, is one of the most widely grown and consumed vegetables worldwide and is used as a traditional medicine in Korea and China in the treatment of food dyspeptic retention, constipation, diarrhea, dysentery and panting. RS has various pharmacological activities, such as alcoholic fatty liver disease amelioration, anti-inflammatory and antitumor effects [25,26]. *A. hookeri* and *R. sativus* L. contain sulfur components, which convert toxic fat-soluble substances into water-soluble substances in the human body, as well as bind toxic substances with sulfur and excrete them via bile or the kidneys [27,28,29].

In this study, we established an optimal ratio of mixture using extracts prepared from AH whole plant, RS and CL roots, AS stems and DM leaves and stems via bioassay-guided selection. As a result, we produced nutraceuticals with anti-inflammatory effects for asthma.

## 2. Materials and Methods

### 2.1. Samples

#### 2.1.1. Extract Preparation

The plants used in this study, RS and CM roots, AH whole plant, AS stems and DM leaves, were purchased from Jeonnam Herbal Medicine Farmer’s Cooperative (Jeonnam, Korea). The dried and powdered samples were added to 10 volumes of distilled water and extracted at 100 °C for 5 h. The extracts were filtered, concentrated at 78 °C using an evaporator (N-1300EW; EYELA, Tokyo, Japan) in vacuo and stored at −70 °C.

#### 2.1.2. Extraction and Analysis of Nutmeg Essential Oils

Essential oils of nutmeg seeds were used to remove the sulfur flavor of the extracts. Nutmeg seed powder was added to 10 volumes of DW and extracted at 100 °C for 4 h to obtain essential oils. The solution was treated with sodium sulfate to remove all water particles and then filtered to extract the essential oils. The essential oils were analyzed using gas chromatography/mass spectrometry (GC/MS; Shimadzu QP2020 Headspace-GC, Kyoto, Japan) with an RTx-5MS Fused-Silica Capillary Column (30 m × 0.25 mm ID; J&W Scientific, Folsom, CA, USA). The oven temperature was maintained at 60 °C for 5 min and then at 250 °C for 10 min using carrier N_2_ gas.

#### 2.1.3. Product Preparation

The powder prepared from RS and CL roots, AH whole plant, AS and DM stems and DM leaves was added to 10 volumes of water, mixed, and extracted under reflux at 100 °C for 5 h. The crude extract was filtered using filter paper (Whatman, Merck, Darmstadt, Germany), evaporated in vacuo (N-1300EW, EYELA, Tokyo, Japan) and freeze-dried. The material was prepared by mixing freeze-dried powder according to the ratio M4, which has potent anti-inflammatory activity. Granular product (GP) was manufactured by mixing the material with additives, i.e., lactose, oligosaccharide, stevioside extract and nutmeg seed essential oil, in a ratio of 1:4 using a granulation machine. Finally, the granules were dried and packaged.

### 2.2. In Vitro Antioxidant Activity

Antioxidant activity of the samples was examined using 2,2-azino-bis (3-ethylbenzothiazoline-6-sulfonic acid) diammonium salt (ABTS) and 1,1-diphenyl-β-picrylhydrazine (DPPH) radical assays. The ABTS radical assay was performed in accordance with the ABTS cation decolorization assay, with slight modifications [30]. ABTS (7 mM; Sigma-Aldrich, St. Louis, MI, USA) was mixed with 2.45 mM potassium persulfate and reacted for 16 h at 4 °C. Typically, sample (50 µL) and ABTS solution (100 μL) were mixed and allowed to react at 23 °C for 20 min, and absorbance was measured at 732 nm using a microplate spectrophotometer (Epoch, Biotek Instruments, Inc., Winooski, VT, USA). The DPPH radical assay was performed in accordance with a previously reported, slightly modified method [31]. Briefly, an equal volume of DPPH solution (0.2 mM) and the samples were mixed in a 96 well plate for 5 s. The mixture was reacted for 30 min in the dark, and absorbance was measured at 517 nm. Garlic acid was used as the positive control of ABTS and DPPH radical assays.

### 2.3. In Vitro Hepatoprotective and Anti-Inflammatory Studies

#### 2.3.1. Cell Culture

Murine-derived macrophage (RAW 264.7) and hepatoma (hepa-1c1c7) cells were obtained from the Korean Cell Line Bank (KCLB; Seoul, Korea) and were cultured in Dulbecco’s Modified Eagle Medium (DMEM: HyClone, Logan, UT, USA) and Minimum Essential Medium (MEM; HyClone), respectively. These cells were cultured at 37 °C in a medium supplemented with 10% fetal bovine serum (FBS; HyClone), 100 U/mL penicillin and 100 μg/mL streptomycin (HyClone) in a humidified atmosphere containing 95% air and 5% CO_2_.

#### 2.3.2. Cell Viability Assay

The cytotoxicity of the samples was assessed using an 3-(4,5-dimethylthiazol-2-yl)-5-(3-carboxymethoxyphenyl)-2-(4-sulfophenyl)-2H-tetrazolium (MTS) assay. The cells (1 × 10^4^ cells/well) were seeded and incubated for 24 h in a medium supplemented with 10% FBS. Next, the culture medium was removed, and the cells, in a serum-free medium, were treated with various concentrations of the extract. After treatment for 24 h, the cells were treated with EZ-CYTOX (DoGEN, Seoul, Korea) for 1 h, and absorbance was measured at 450 nm (Epoch).

#### 2.3.3. Hepatoprotective Effects

The hepatoprotective effect of all extracts was investigated by measuring glutathione S-transferase (GST) activity using a spectrophotometric assay [32]. Hepa-lclc7 cells (1 × 10^4^ cells/well) were treated in a serum-free medium for 48 h, lysed with 2% Triton X-100 and mixed for 10 min. The lysate was mixed with 100 μL of 2.5 mM GSH in 0.1 M potassium phosphate buffer and 1 mM 1-chloro-2,4-dinitrobenzene (DCDNB) for 1 min, and absorbance was measured at 405 nm for 5 min. GST activity was calculated as slope/min/mg protein and expressed as the ratio of GST activity of the cells treated with extracts to that of the control.

#### 2.3.4. Anti-Inflammatory Effects

##### Nitric Oxide Production Assay

RAW 264.7 cells (1 × 10^5^ cells/well) were treated in a serum-free medium for 1 h and then stimulated with lipopolysaccharide (LPS; 1 µg/mL). After 16 h, the supernatant and equal volumes of Griess reagent A (0.1% (*w*/*v*) N-(1-naphthyl) ethylenediamine in DW) and B (1% (*w*/*v*) sulfanilamide in 5% (*v*/*v*) phosphoric acid) were added and incubated at room temperature (RT) for 15 min, and the absorbance was measured at 540 nm. Nitric oxide (NO) production was measured using a serum-free culture medium as a control.

##### Cytokines Generation Assay

Cytokine generation in LPS-induced RAW 264.7 cells was measured using enzyme-linked immunosorbent assay (ELISA) kits (BD OptEIATM, San Diego, CA, USA) according to the manufacturer’s instructions.

### 2.4. In Vivo Anti-Inflammatory Studies

#### 2.4.1. Animals

Male Sprague-Dawley (SD) rats (160–190 g each) were used to determine the anti-inflammatory effects of M4 and products prepared with M4. The animals were housed (3–5 per cage) in a laminar-air-flow cabinet maintained at 23 ± 1.5 °C, a humidity of 55 ± 15%, and lighting that followed a 12 h on/12 h off cycle. The rats were provided a laboratory diet and tap water ad libitum during the experimental period. All animals were approved by the Institutional Animal Care and Use Committee of Dongshin University (No. DSU2016-10-01).

#### 2.4.2. Sensitization and Inhalation Exposure

A schematic representation of sensitization and inhalation exposure is shown in Figure 1. Rats were divided into six groups: group 1 (non-treated normal (NOR)), group 2 (ovalbumin (OVA) 1 mg/kg and LPS 5 mg/kg), control (CON)), group 3 ((dexamethasone (DEX) 0.5 mg/kg; (positive control), DEX), and groups 4, 5 and 6 (sample concentrations at 100, 200, and 400 mg/kg, respectively). To induce allergic asthma, rats in groups 2–6 were sensitized twice with an intraperitoneal (i.p.) injection of OVA (1 mg/kg), emulsified in aluminum potassium sulfate in phosphate-buffered saline (PBS), and LPS (5 mg/kg) every five days. Nebulized 2% PM (Standard Reference Matter, NIST, Gaithersburg, MD, USA), 3% OVA and 5% LPS in PBS were administered by inhalation for 15 min at 4 h intervals for a total of four times on days 6, 7, 9 and 11 after i.p. injection.

#### 2.4.3. Real-Time Reverse Transcription-Polymerase Chain Reaction (RT-PCR)

The gene levels of OVA-, LPS- and PM-induced cytokine generation were assessed using real-time RT-PCR with specific primers (Table 1). Total RNA was extracted from OVA-, LPS- and PM-treated lung tissue using an extraction kit (TRIzol) according to the manufacturer’s protocol. For complementary DNA (cDNA) synthesis, 5 μg of total RNA was reverse-transcribed using oligo dT, (2.5 μL) and RT premix (Bioneer, Daejeon, Korea). cDNA samples (50 μL) were amplified using RT-PCR with specific primers (Table 1). Glyceraldehyde-3-phosphate dehydrogenase (GAPDH) was used as the internal control. RT-PCR conditions were as follows: pre-denaturation at 94 °C for 5 min, 30 cycles of denaturation at 94 °C for 30 s, annealing at 55–60 °C for 30 s and extension at 72 °C for 1 min.

#### 2.4.4. Cytokines Generation Assay

Proinflammatory cytokines, namely interleukin (IL)-1β, IL-6, and TNF-α produced by OVA-, LPS- and PM-stimulated rats were determined using an ELISA kit (BIOMATIK, Ontario, Canada) and Invitrogen (Thermo Scientific, Waltham, MA, USA) according to the manufacturer’s protocol. Cytokine generation in LPS-induced RAW 264.7 cells was measured using ELISA kits (Invitrogen; Thermo Scientific) according to the manufacturer’s instructions.

### 2.5. Statistical Analysis

Comparisons were made through one-way analysis of variance (ANOVA) and Duncan’s multiple range test using IBM SPSS Statistics 26 (Chicago, IL, USA). Values are reported as the mean ± standard error (SE), and statistical significance was set at *p* < 0.05.

## 3. Results

### 3.1. In Vitro Antioxidant and Hepatoprotective Effects of Mixture

Herbal plants, namely RS, AH, AS, DM and CL, were selected for the preparation of functional foods to treat inflammatory disorders caused by asthma. As shown in Table 2, these plants were extracted with water and mixed in various ratios. Antioxidant activity has been reported to affect various bioactivities such as skin whitening and anti-inflammation [33]. ABTS and DPPH radical scavenging actions offer redox-functionalized proton ions for unstable free radicals and play a critical role in stabilizing detrimental free radicals; these assays were conducted to assess the antioxidant activity of each mixture [31,32]. The IC_50_ values of M1, M2, M3 and M4 for ABTS radical scavenging activity were 214, 61, 150 and 57 (μg/mL), respectively (Figure 2A). M2, M3 and M4 showed strong ABTS radical scavenging activity (Figure 2A). The IC_50_ values of M1, M2, M3 and M4 for DPPH radical scavenging activity were 321, 171, 209 and 197 μg/mL, respectively (Figure 2A). Among the mixture, M2 and M4 exhibited potent DPPH radical scavenging activity (Figure 2A). To examine the hepatoprotective effects, Hepa-lc1c7 cells were treated with a mixture of extracts at 50, 100 and 200 μg/mL, and GST activity was investigated. As shown in Figure 2B, the GST activity of M4 was 4.09 ± 0.58, which was 1.78 times higher than that of M1 (2.77 ± 0.44).

### 3.2. Anti-Inflammatory Effects of Mixture

To investigate anti-inflammatory effects, the levels of NO and proinflammatory cytokines were examined in LPS-induced murine macrophages. All mixtures inhibited LPS-induced NO production without causing cytotoxicity (Figure 3A). Specifically, M4 exhibited the strongest inhibition of LPS-induced NO production in a dose-dependent manner (Figure 3A). To confirm the anti-inflammatory effects of the mixture, the levels of proinflammatory cytokines were measured using an ELISA kit. Compared with the untreated control group, the levels of LPS-induced proinflammatory cytokines IL-1β, IL-6 and TNF-α increased. When macrophages were pretreated with the extracts, we observed extract-mediated downregulation of proinflammatory activation at 10, 25, 50 and 100 μg/mL (Figure 3B). In particular, M4 exhibited the strongest inhibitory activity against proinflammatory cytokines compared with the other mixture groups, and hence was prepared based on bioassay-guided selection. Based on these results, M4 with the most potent anti-inflammatory and antioxidant effects was selected for the preparation of products to protect against inflammatory disorders.

### 3.3. Antiasthmatic Effects of M4

The antiasthmatic effect of M4 was assessed by measuring the proinflammatory cytokine levels in OVA-, LPS- and PM-induced rat models. The gene expression of proinflammatory cytokines in the lungs of OVA-, LPS- and PM-treated rats was determined. RT-PCR analysis revealed that treatment with M4 modulated the gene expression of proinflammatory cytokines, such as IL-1β, IL-6 and TNF-α in a dose-dependent manner (Figure 4A). Bronchoalveolar lavage fluid (BALF) was collected from the lungs to investigate the generation of inflammatory cytokines. As shown in Figure 4B, the levels of inflammatory cytokines (IL-1β, IL-6 and TNF-α) in the BAL of the OVA-, LPS- and PM-treated rats were higher than those in the PBS-treated group. However, the M4-treated group showed significantly reduced levels of these cytokines in a dose-dependent manner (Figure 4B).

### 3.4. Sulfur Flavor-Reduction Effects Using Essential Oil of Nutmeg Seed

To reduce sulfur flavor in the M4, essential oils of nutmeg and cinnamon were used. As shown in Figure 5A, the essential oil content of nutmeg seed and cinnamon bark was 11.5 mL/kg and 9.5 mL/kg, respectively. Nutmeg seeds contain many essential oils, and their components have been analyzed. The essential oils of nutmeg seed included 15 flavor components, with alpha-pinene, sabinene and beta-pinene as the major components (Figure 5B). Therefore, the essential oils of nutmeg seeds were added to the product using M4 to remove the sulfur flavor.

### 3.5. Anti-Inflammatory Effects of Product

Based on these results, GP was manufactured using adjusted additives such as oligosaccharides, steviosides, and essential oils of nutmeg seeds as their main ingredients. The anti-inflammatory activity of the product was assessed in the BALF of the lungs from OVA-, LPS-, and PM-induced rat models. As shown in Figure 6, the levels of the proinflammatory cytokines IL-1β, IL-6 and TNF-α in the BALF of OVA-, LPS- and PM-stimulated rats were higher than those in the PBS-stimulated group. However, the GP-treated group showed reduced levels of IL-1β, IL-6 and TNF-α in a dose-dependent manner (Figure 6).

## 4. Discussion

Asthma is a chronic and complex airway disease that affects more than 350 million people worldwide and is heavily influenced by diverse lifestyles, genetics and environmental factors [6]. Allergic asthma is characterized by type 2 airway inflammation. Ambient air pollution caused by many industries has raised great concern worldwide [8]. PM is an extrinsic factor that can trigger inflammation, a set of complicated immunovascular reactions designed to protect the host against external or internal stimuli [9,12]. Asthma medications include anti-inflammatory drugs and bronchodilators; however, they cause side effects such as depression, agitation and sleep disturbance [1]. Hence, there is an urgent need to develop nutraceuticals that can prevent adverse effects caused by drugs and PM-induced inflammatory diseases such as asthma that affect the human body. Natural products have attracted extensive attention owing to their usefulness in drug discovery and functional food production. The combination of herbal medicines is effective in the treatment and prevention of various diseases owing to synergistic effects [13,14]. In the present study, we assessed the optimal ratio of herbal plant extracts that produced anti-inflammatory effects for asthma treatment using a bioassay-guided selection. The herbal plants RS, AH, AS, DM and CL were selected, and mixtures of various ratios were prepared (Table 2). Among the mixtures, M2 and M4 exhibited strong scavenging activities of ABTS and DPPH radicals (Figure 2A). GST catalyzes the conjugation of the reduced form of GSH to xenobiotic substrates for detoxification [32]. To assess the hepatoprotective effects of the mixture, GST activity was examined in murine hepatocytes (hepa-1c1c7). Among the mixtures, M4 exhibited the strongest GST activity (Figure 2B).

Inflammation is the response of the immune system to hazardous stimuli such as pathogens, damaged cells, toxic compounds, inflammatory cytokines or irradiation. It acts by eliminating injurious stimuli and initiating the healing process; therefore, it is a defense mechanism that is vital to health. Macrophages are major inflammatory and immune effector cells, and their activation is triggered by exposure to bacterial LPS or proinflammatory cytokines. Activated macrophages produce inflammatory mediators such as NO and proinflammatory cytokines such as IL-1β, IL-6 and TNF-α. The excessive production of these molecules can lead to chronic inflammatory diseases. Our results showed that M4 most strongly inhibited NO, IL-1β, IL-6 and TNF-α generation without causing cytotoxicity (Figure 3A,B).

Allergic asthma, triggered by various environmental factors such as air pollutants, is a prolonged lung inflammatory disorder caused by the infiltration of eosinophils, neutrophils, and leukocytes into the bronchoalveolar space [1,8,9]. Furthermore, proinflammatory cytokines enhance inflammatory diseases, such as asthma and chronic obstructive pulmonary disease [34]. We examined the anti-inflammatory effects of M4 on asthma using OVA, -LPS- and PM-induced rat models. Our results showed that M4 inhibited the gene expression and generation of the proinflammatory cytokines, IL-1β, IL-6 and TNF-α (Figure 4A,B). AH, belonging to the *Allium* family, has a strong sulfur flavor; hence, essential oils of nutmeg seed, which contains a higher essential oil content than cinnamon bark, were added to reduce its flavor (Figure 5A). Our results showed that the essential oil of nutmeg seed had 15 types of flavor components, with alpha-pinene, sabinene and beta-pinene as major components (Figure 5B), which are also found to exhibit antiallergic effects when extracted from *Lepechinia betonicifolia* and *Gynoxys miniphylla* [35,36,37]. Major components were detected in essential oils from *Flos magnoliae* with inhibitory effects on allergic rhinitis [38,39]. We suggest that the addition of nutmeg seed essential oils effectively reduces the sulfur flavor. Based on these results, GP was prepared using M4 supplemented with the essential oils of nutmeg seeds according to the food manufacturing process. Cytokines play a significant role in asthma and promote the survival of various inflammatory cells [1,34]. Proinflammatory cytokines such as IL-1β, IL-6 and TNF-α are found in increased amounts in the sputum and BALF of patients with asthma [34]. Our results demonstrated that GP inhibited the generation of IL-1β, IL-6 and TNF-α in OVA-, LPS- and PM-induced asthmatic animal models (Figure 6). Proinflammatory cytokines amplify inflammation through the activation of NF-κB, leading to the increased expression of inflammatory genes [1,2,3,4,34,40]. Overall, the results of our study indicate, for the first time, that the product prepared using an herbal mixture suppresses inflammatory responses in diseases such as asthma (Figure 7). Further studies on the regulatory action of products in inflammatory signaling are needed to ensure their potential therapeutic application in the treatment of inflammatory disease including asthma. Taken together, our results suggest that functional foods containing a mixture of herbal plants can be a potential candidate for the treatment of inflammatory diseases.

## 5. Conclusions

Four types of mixtures were prepared using five kinds of herbal plant extracts: RS and CM roots, AH whole plant, AS stems and DM leaves. M4 exhibited the strongest increase in GST activity and anti-inflammatory activity in asthma and thus was selected to prepare the product. GP was manufactured using M4 and additives including nutmeg seed essential oils. GP inhibited the generation of proinflammatory cytokines IL-1β, IL-6 and TNF-α in OVA-, LPS- and PM-induced asthmatic rat models. These results suggest that GP prepared with M4 is a potent functional food with anti-inflammatory activity that can be used to treat inflammatory diseases such as asthma caused by ambient air pollutants.

## Figures and Tables

**Figure 1 foods-11-01910-f001:**
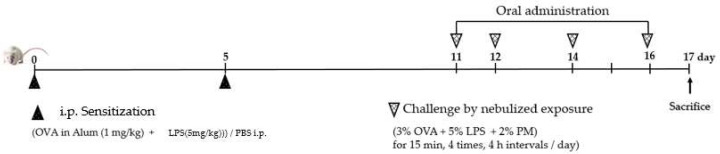
Experimental design for rat model of allergic asthma. SD rats were sensitized with an i.p. injection of OVA-, and LPS- on days 0 and 5 nebulized OVA-, LPS-, and PM were administered by inhalation of nebulized on days 11, 12, 14, and 16. Oral administration, consisting of DEX or each mixture, was performed from day 11 to day 16 of the protocol.

**Figure 2 foods-11-01910-f002:**
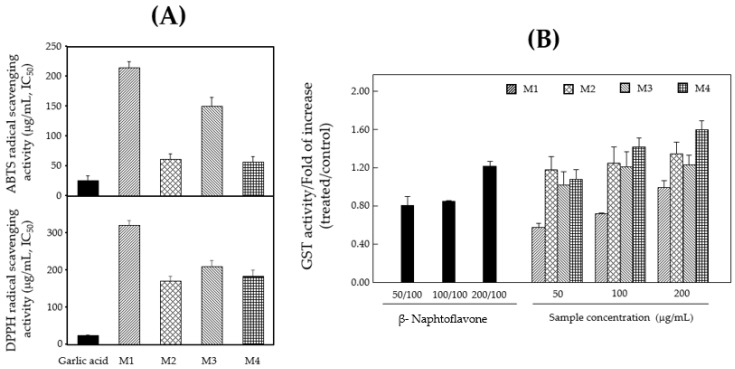
Antioxidant and hepatoprotective effects of each mixture. (**A**) ABTS and DPPH radical scavenging activities. (**B**) GST activity in hepa-lclc7 cells. Data are expressed as means ± SD (*n* = 3) of three individual experiments. A value of *p*-value < 0.05 was considered statistically significant.

**Figure 3 foods-11-01910-f003:**
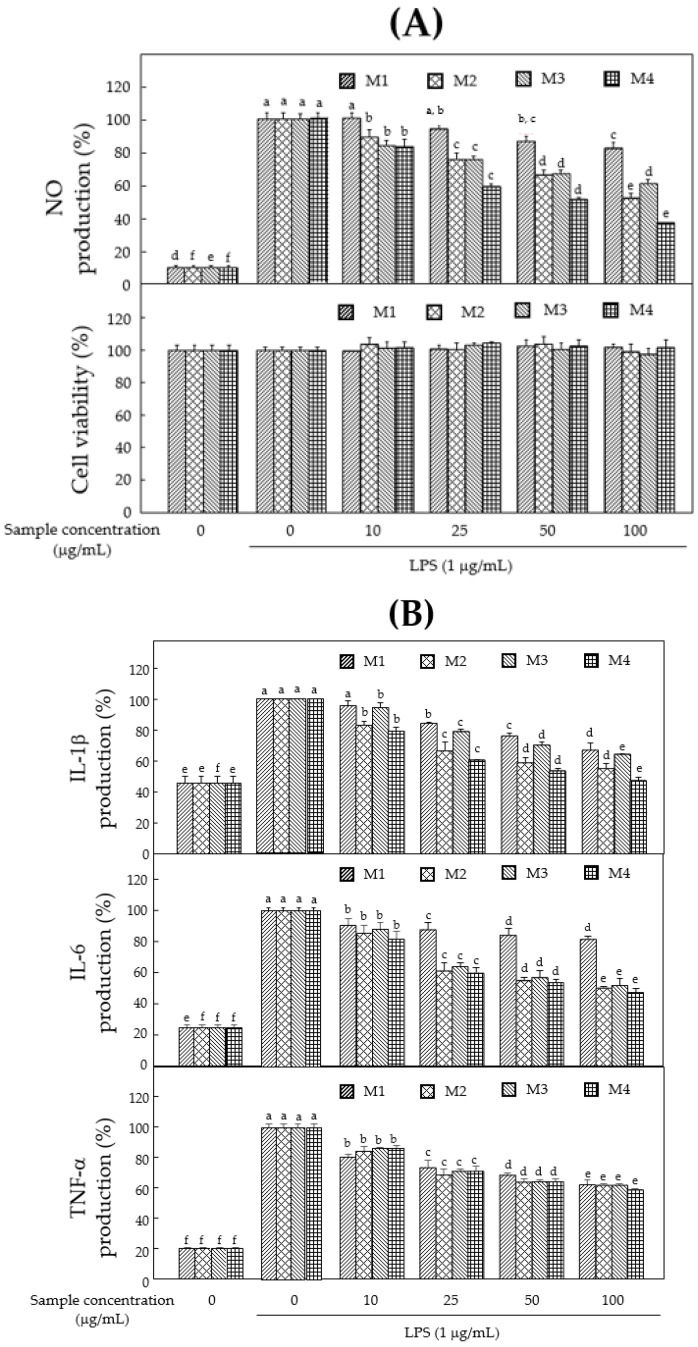
Anti-inflammatory effects of each mixture in LPS-induced murine macrophage cells. (**A**) NO production and cytotoxicity. (**B**) Generation of proinflammatory cytokines, IL-1β, IL-6 and TNF-α in LPS-stimulated RAW 264.7 cells. Data are expressed as the means ± SD (*n* = 3) of three individual experiments. Different letters of *p*-value < 0.05 were considered statistically significant.

**Figure 4 foods-11-01910-f004:**
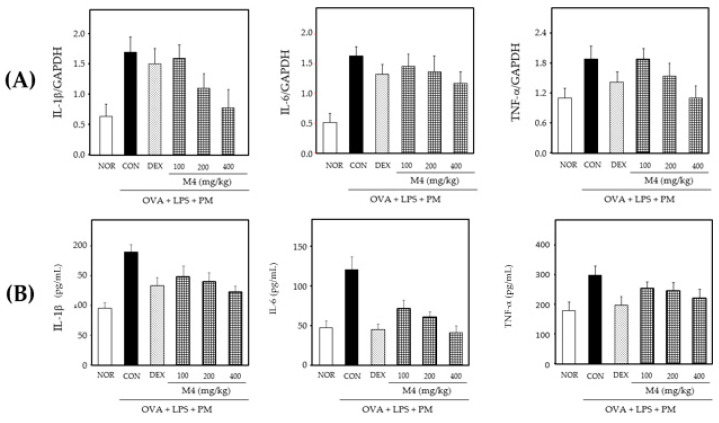
Anti-inflammatory effects of M4 in OVA-, LPS- and PM-induced asthmatic rat model. (**A**) Gene expression and (**B**) generation of proinflammatory cytokines IL-1β, IL-6 and TNF-α in OVA-and LPS-challenged and OVA-, LPS- and PM-sensitized asthmatic SD rats. Data are expressed as the means ± SD (*n* = 5). A value of *p*-value < 0.05 was considered statistically significant.

**Figure 5 foods-11-01910-f005:**
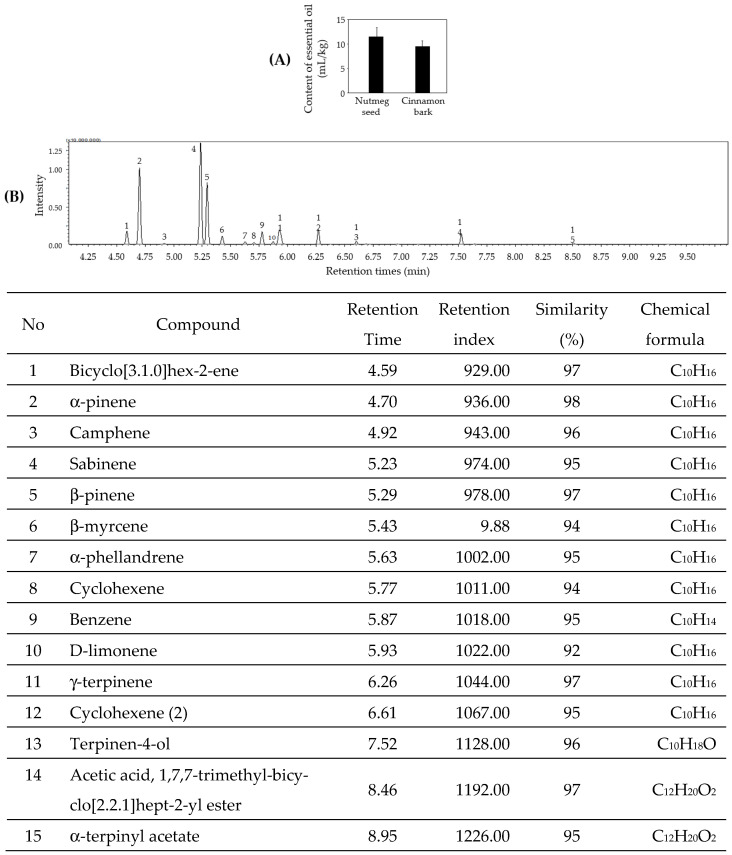
Nutmeg seeds essential oils content and components. (**A**) Essential oil content of nutmeg seed and cinnamon bark. (**B**) Essential oil component of nutmeg seed analyzed by GC-MS.

**Figure 6 foods-11-01910-f006:**
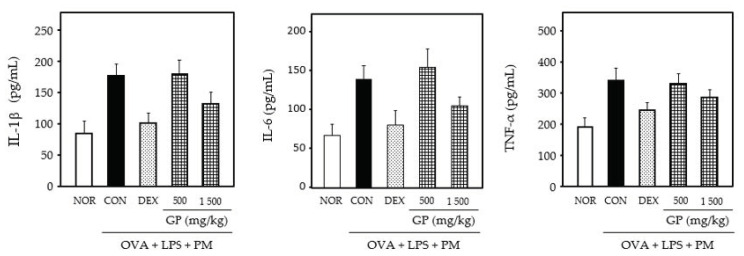
Anti-inflammatory effects of the GP. Generation of proinflammatory cytokines such as IL-1β, IL-6 and TNF-α in OVA-, LPS- and PM-induced asthmatic rat model. Data are expressed as the means ± SD (*n* = 5). A value of *p*-value < 0.05 was considered statistically significant.

**Figure 7 foods-11-01910-f007:**
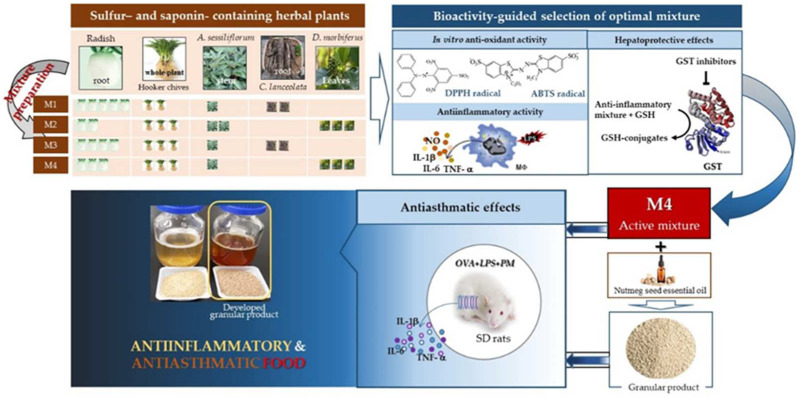
Schematic representation of anti-inflammatory food prepared with mixture of herbal plants and its use in treating asthma.

**Table 1 foods-11-01910-t001:** Primer sequence used in this study.

Primer	Forward	Reverse
IL-1β	5′-ATGGCAACTGTTCCTGAACTCAACT-3′	5′-CAGGACAGGTATAGATTCTTTCCTTT-3′
IL-6	5′-TTGCCTTCTTGGGACTGATG-3′	5′-CAGAATTGCCATTGCACAACT-3′
TNF-α	5′-CCACATCTCCCTCCAGAAAA-3′	5′-AGGGTCTGGGCCATAGAACT-3′
GAPDH	5′-AACGGCACAGTCATGGCTGA-3′	5′-ACGCCAGTAGACTGCACGACAT-3′

**Table 2 foods-11-01910-t002:** Optimal composition of SSM.

Sample Name	Source	Ratio of Mixture
M1	RS root:AH whole plant:AS stem:CL root	50:20:10:20
M2	RS root:AH whole plant:AS stem:DM root and leaves	20:30:20:30
M3	RS root:AH whole plant:AS stem:CL root	40:30:10:20
M4	RS root:AH whole plant:AS stem:DM root and leaves	30:30:10:30

## Data Availability

The data presented in this study are available on request from the corresponding author.

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
