# Peer review of "Anti-Inflammatory Food in Asthma Prepared from Combination of Raphanus sativus L., Allium hookeri, Acanthopanax sessiliflorum, and Dendropanax morbiferus Extracts via Bioassay-Guided Selection"

_foods, 2022, doi:10.3390/foods11131910_

Round 1

Reviewer 1 Report

The authors investigated the effects of herbal plants mixture to reduce inflammation and treat asthma in rats. However, the experimental designs appear unclear; and it is still an incomplete study due to lacked investigation on the underlying mechanism.   

Here are recommendations:

  1. The authors suggested that sulfur- and saponin contents play the major roles to ameliorate inflammation and asthma. Why don’t they examine the effects of total sulfur and saponin? The whole extract, especially for the herbal plants mixture in current study contain various ingredients. It is complicated to tell which bioactive ingredients are crucial to exert protective effects. And the quality control and molecular fingerprint information for the extracts are lacking.
  2. There is no investigation on the underlying mechanism for the anti-inflammatory and antioxidant activities. The authors mostly studied about the anti-inflammatory effects. They did not show much findings related to asthma. The authors studied the antioxidant activities which should be further explored in the asthma rat model.
  3. Please check the English proficiency. There are grammatical mistakes and errors in expressions. I cannot understand the title and running title which should be revised. And followings are some examples which should be revised.
  • “Saponins are secondary compounds widely distributed in many plants, and form a stable foam in aqueous solutions such as soap, hence, the name “saponin”
  • “It is composed of gaseous constituents and airborne particulate matter (PM), which is classified based on particle size (known as the aerodynamic equivalent diameter), as smaller than 10 μM, 2.5 μM, or ultrafine size”
  1. In abstract, the full forms should be mentioned when abbreviations are first used, i.e. ABTS, DPPH, GST, NO, IL-1β , IL-6 and TNF-α; and abbreviations can be used afterwards, i.e. OVA, LPS and PM.
  2. In introduction, the authors should justify the purpose of the research. It should also indicate the novelty of the research. Why were the sulfur- and saponin containing herbal plants namely radish root, whole plant of hooker chives, stems of Acanthopanax sessiliflorum, leaves of Dendropanax morbiferus, root of Codonopsis lanceolate selected for investigation in this study? This is not described in section 3.1 either, only various searches using NDSL, Google and NCBI are mentioned.
  3. For section 2.1.1, why the herbal plants were extracted by that method? Reference should be provided.
  4. What is MTS assay in section 2.3.2.?
  5. “Fig. ????” in section 2.4.2, and “Fig. ??” in the Discussion in page 9 need to be revised.
  6. In discussion, “The pattern of active materials of their water extract were investigated the peaks in the chromatogram by LS-MS/MS, and the active substance was estimated to be a compound of the terpenoids series (data not shown)”. This data should be shown.
  7. In Figure 1, control is lacking.
  8. There are no indication of statistical significance in figure 1. And what are the labels a, b, c, d, e, f stand for in Figures 2, 3?
  9. For animal study, n=3 for experiments in rats is not enough. More repeats are needed.

Author Response

Attached file.

Reviewer 2 Report

In this article, an optimal mixture of sulfur- and saponin-containing herbal plants was selected to produced anti-inflammatory foods for asthma treatment by comparing antioxidant activities, GST activity and ability to inhibit the production of inflammatory mediators. Then this product has been shown to inhibit production of proinflammatory cytokines in asthmatic rat model. Some comments are offered for author’s consideration.

Abstract:

1.There are grammar and spelling errors occurred in the abstract such as “Asthma is very common inflammatory diseases” and “polluants”.

2.There are format errors occurred in the abstract. It is supposed to write nouns in full explanation before abbreviating. For example, “GST”.

3.It is better to use abbreviations that have been explained before to ensure brevity in the manuscript.

Introduction:

Grammar errors: “hence, the name saponin”, “Ambient air pollution is major environmental threat” and other grammatical errors due to incorrect use of conjunctions.

Materials and Methods:

1.In chapter 2.2, the positive control gallic acid should be supplemented with abbreviation because it appeared in the figure.

2.In chapter 2.2.2, the number of the figure is not indicated. The same problem exists in the conclusion.

Results:

1.Almost all of the number of the figures do not match the instructions in the text. In addition, some instructions of the figures that should have appeared in the text were missing.

2.In chapter 3.4, the explanation of CST activity was wrong according to the figure, and there is no explanation of “β-naphtoflavone” in the Fig. 1 (B).

3.No explanation of the use of “PLA” in the Fig. 3.

Discussion:

1.The number of the figures do not match the instructions in the text.

2.No explanation of “GG” was given.

Conclusion:

The research confirming the result that the product is a potent therapeutic food for the treatment of inflammatory disease in asthma caused by ambient air pollutants is not very sufficient. It is better to add more experimental testing to strengthen the findings.

Others:

1.Authors should correct carefully the typing and grammatical errors throughout this manuscript. The language should be improved and more details should be noted.

2.Spelling error in “Figure Legends”: Figure 4 and 6 “Antinflammatory”.

Author Response

Attached file

Reviewer 3 Report

The subject is interesting and the work provides valuable information, but the article writing and typing needs to be checked carefully.

Specific comments

1.    The running title seems to be missing the word "from".

2.    The Abstract section is poorly written, it should contain enough quantitative information that will allow a reader to understand the key findings of the research. In addition, the plant name that appear the first time should provide the full name, not the abbreviated name.

3.    Section 2.1.1: The extract preparation has no drying step, how to calculate the subsequent concentration? Did they use wet weight?

4.    Section 2.1.3: How the authors calculate the crude saponin content, weight (dry or wet), spectrophotometry or others?

5.    Section 2.4.2: Figure 4 should be moved here and renamed to Figure 1.

6.    Section 3.1: The authors should explain why they chose the four herbal plants and how they selected the compositions of the four mixture in Table 3.

7.    Figure 5 and 6 should be exchanged.

8.    Section 3.1: Figure number should be checked. Figure 5B should show the relative contents of each compound.

9.    Figure 6: What are “PEN” and ”DEX” should be stated in the figure caption.

10.      The reference section should be modified carefully to meet the journal’s format.

Author Response

Attached file.

Round 2

Reviewer 1 Report

Although the authors have made some changes to the manuscript, there has not been a substantial improvement to the quality of the manuscript. To truly reflect the title of the manuscript, the effects of the extract against asthma should be investigated in detail. Inflammation and oxidative stress is associated with many diseases, not only asthma. More direct evidence for the protective effects against asthma should be provided. For instance, increasing rate of specific airway resistance and decreasing rate of dynamic compliance of asthmatic rats; sectioning of the lung stained with H&E. There is no mechanism to show how the extract suppress inflammation and oxidative stress. The whole extract, especially for the herbal plants mixture in current study contain various ingredients. It is complicated to tell which bioactive ingredients are crucial to exert protective effects.  

Author Response

File attached.

Reviewer 2 Report

I think this manuscript could be accepted in its current form.

Author Response

File attached.
